# Impact of the Product Master Data Quality on the Logistics Process Performance

Diana Božić [1,*], Margareta Živičnjak [1], Ratko Stanković [1] and Andrej Ignjatić [2]

[1] Faculty of Transport and Traffic Science, University of Zagreb, Vukelićeva 4, 10000 Zagreb, Croatia; mzivicnjak@fpz.unizg.hr (M.Ž.); rstankovic@fpz.unizg.hr (R.S.)

[2] Faculty of Organization and Informatics, University of Zagreb, Pavlinska 2, 42000 Varaždin, Croatia; anignjatic@student.foi.hr

[*] Correspondence: dbozic@fpz.unizg.hr

**Abstract:** *Background:* The importance of up-to-date product master data in the digital age should not be underestimated. However, companies still struggle to ensure high-quality product data, especially in the field of logistics. Hence, the focus of our research lies in the disregard of the importance of product data quality to the performance of logistics processes. *Methods:* The analysis of the influence of product data on the performance of logistics processes was carried out using data from two fast-moving consumer goods (FMCG) distribution and retail companies. Data were gathered via interviews, while process activities were timed using a stopwatch, and interruptions were documented. The significance of the impact was determined using inferential statistical procedures based on the variable and the measurement scale type employed. *Results:* The quality of product master data has a significant impact on the performance of logistics processes; while managers are aware of the complications, they lack the motivation to detect and analyse such inaccuracies. *Conclusions:* The findings enhance comprehension of the obstacles generated by inadequate product data in logistics, which obstruct optimisation, and offer numerical proof of the impact of product data quality on logistics performance, thus expanding the current body of research.

**Keywords:** product master data; logistics process performance; fast-moving consumer goods; supply chain

## 1. Introduction

The data that identify and describe a product are used throughout the supply chain, from logistic operations to the shop floors, web shops, printed catalogues, marketing, invoices, data pools, or claiming guarantees. Even small organisations may deal with thousands of products whose data need to be managed. Therefore, managing the product master data becomes essential for everyone involved in the supply chain.

The product master data represent a unique identification mark in data exchange between supply chain stakeholders. In the literature, it is also known as the product information—PI [1], which describes its function, features, usage, and configuration. It can involve different attributes, relationships, and records, logged in separate systems and locations in the form of numbers, text, structure, relationships, and assets such as images, videos, and documents. In the context of this research, the features of accuracy, completeness, and timeliness are considered the product master data quality. According to [2], in the fast-moving consumer goods industries (FMCG), PI involves the Global Trade Identification Number (GTIN) for identification of the product and its manufacturer, brand name, respective market, technical characteristics of the product, such as height, depth, width, weight, associated components, features relevant for logistic activities (dimensions of the transport box or pallet to be used for transport and storage) and for marketing purposes (availability, packaging design, etc.).

Access to product information and traceability of its movements in the supply chain has proven to be a crucial criterion for the quality and safety of a product [3]. The most accurate and comprehensive information on the product can be provided by its manufacturer. By applying proper product labels and protocols for data exchange, manufacturers can ensure the accuracy of the product data flow through the supply chain. According to the recent research [2,4–9], this issue still has not been widely addressed.

Data may be stored in spreadsheets or single-product databases during the product life cycle, across separate systems, departments, and entities, which incurs a possibility of having several different versions of data for a single product. Incorrect product descriptions can lead to product returns and customer complaints, lower efficiency of supply chain operations, and additional work [10]. Inaccurate or insufficient product data negatively impacts the product itself and all processes related, causing disturbances in the supply chain, such as out-of-stock situations [11], although they may also occur due to inadequate delivery systems.

In the information age, product data are considered an integral part of the product. And yet, many companies still struggle to ensure product data accuracy and availability across their processes, systems, and the whole supply chain [2,6,10,12–14].

The origins of product information can vary, originating either from the manufacturer or the supplier. Once the product is uniquely identified and ready for delivery to the end user, it must be recorded in the retailer and/or logistics company database. To be included in the respective database, the product master data must be exchanged between the business partners. It is at this stage that most problems arise in connection with the product master data [15]. While research in the realm of digitalisation and data synchronisation in the supply chain posits that product data are an unquestionable actuality [4,5,7,16–21], or implies that they are necessary [18,19,22,23], one might be inclined to deduce that precise and punctual product data are an indisputable truth, thus rendering further extensive research unnecessary. Recent studies reveal that the quality of product data is pivotal for improving performance through the digitalisation process [24], and digitalisation is indispensable for the survival of organisations in the market [25,26]. However, the practicalities of logistic operations present a contrasting perspective.

The simplification and optimisation of logistics activities through technological development and the diversity of information systems [8,16,17,27] show their weaknesses, while considerable resources are spent on correcting errors in product data when receiving or shipping products, returning products, and the like [2,6,28]. Confirmation of the above statements can be found sporadically in various studies on the realisation of logistics processes.

Through a detailed analysis of published studies on the quality of product master data and industry practice, we identified a gap in the scientific literature on the impact of low-quality product data on the performance of the logistics process. Our study therefore pursues two objectives:

- To assess management awareness of the importance of product data quality for the logistics industry;
- To quantify the influence of inaccuracies in the product data on the performance of logistics processes.

The present study is, so far as we know, the first use of direct process activities observation with the use of a stopwatch and measurement to analyse overall product data quality influence on the operational performance of the logistics process in the real sector.

The rest of this paper is structured as follows. The section that follows this introduction shows a relevant literature review. Section 3 explains product and master data in detail. In Section 4, the methodology for research is proposed. The results of the research are shown in Section 5. Discussion and conclusions are given in Section 6, including relevant implications and future research ideas.

## 2. Literature Review

Studies that emphasise the importance of PI for supply chain optimisation can be divided into those that focus on the collection and management of PI and those that focus on PI exchange in the different parts of the supply chain.

Database architecture [29,30], automatic identification of products (coding structure and content) [31], data quality [10,32,33], data carriers [6,28–34], supply chain digitisation [2,35], and data synchronisation [2,36,37] are the most researched topics around PI collection and management. The abovementioned studies focus on technological solutions and communication platforms and do not target the quality of the exchanged data, which is indispensable for executing logistics processes.

In recent product data studies, two perspectives can be recognised: that of the manufacturer [6,28] and that of the retailer [2,10,21,38]. Retailers are usually the initiators of automatic product data exchange, as they invest a considerable amount of their resources in manually entering product data into their information systems. Such initiatives are not always welcomed by manufacturers, resulting in a lack of logistical product data.

The concentration points of products in any supply chain are warehouses, which tend to be smart and environmentally friendly [39,40]. As research [39] shows, the carbon footprint and associated costs can be optimised by minimising picking time, which is highly dependent on the accuracy and timeliness of data in the warehouse management system (WMS). In terms of data entry in a WMS, the authors [39] note that most WMSs rely on manual data entry without the ability to automatically retrieve or capture data in real time. Since human error is responsible for 80% of erroneous information [40], problems in the execution of warehouse operations are inevitable.

Management awareness of the importance of product data quality and the resulting problems in inter-organisational data exchange are examined in studies [10,20,21]. The survey of 245 German companies [10] shows a high awareness of the importance of product data quality for business efficiency (80%) and of the high costs of ensuring a sufficient level of quality (84%). Despite the high awareness of the importance and associated costs, only 15% of the companies surveyed are familiar with existing methods for improving product data quality (e.g., Global Data Synchronization Network—GDSN), but very few use them (6%). Most of the companies surveyed exchange data with their business partners in non-machine-readable form (Word and PDF documents or MS Excel spreadsheets) and do not check the product master data supplied [10]. This usually leads to a fragmentation of product data within the supply chain. The fragmentation of the supply chain is analysed in a study [20] of the pharmaceutical supply chain in France, where shipments often arrive without documentation, leading to fragmentation of information and thus inefficient data transmission to the receiving company. The quality of product data exchange via the GDSN was analysed by [21] using the example of 22 FMCG retailers in the Netherlands and concluded that the quality of the data still needs to be improved, while the standard itself is not sufficient, especially for the operation of logistics processes.

Common methods for analysing product data quality in supply chains and logistics processes include semi-structured interviews, surveys, literature reviews, and documentation analyses, while the application of quantitative methods has so far only been used in one study [21] to quantify information fragmentation in the supply chain without being focused on logistic performance.

## 3. Demystification of Product Master Data and Logistic Data

Product identification is crucial at every stage of a product's life cycle, including design, production, distribution, and decommissioning, due to the growing need to track product origin and prevent counterfeiting. To meet this requirement, manufacturers or suppliers must assign a unique product identifier and attach a data carrier (also known as an automatic identification device which stores and transmits data relevant to tracking, identification, and management of goods) to the product. Data carriers are utilised to meet different business needs and can store varying quantities of data for different purposes

(commercial, for logistics purpose or handling). They can encode information such as serial numbers, batch/lot numbers, and product attributes [41,42]. Figure 1 illustrates different kinds of data carriers and identification keys they can hold. The DataBar (RSS-14) is a symbology with a predetermined symbol length that is utilised to encode a total of 14 digits, which includes a single-digit indicator with omnidirectional scanning capability at full height or horizontally for smaller markings. EAN/UPC barcodes (EAN-8 holds 8 digits, EAN-13 holds 13 like UPC-A, UPC-E holds 6 digits) are widely recognised for their immediate identification, found on nearly all products worldwide, and are specifically tailored for environments that require frequent scanning, such as the point of sale (POS). ITF-14 is a 14-digit barcode in a symbology known as "Interleaved 2 of 5" and is often used as the primary data carrier for the GTIN-14 data frame. EPCglobal is a standard tailored for industries, designed to streamline the adoption of radio frequency identification (RFID) technology by utilising electronic product codes (EPC) [43–45].

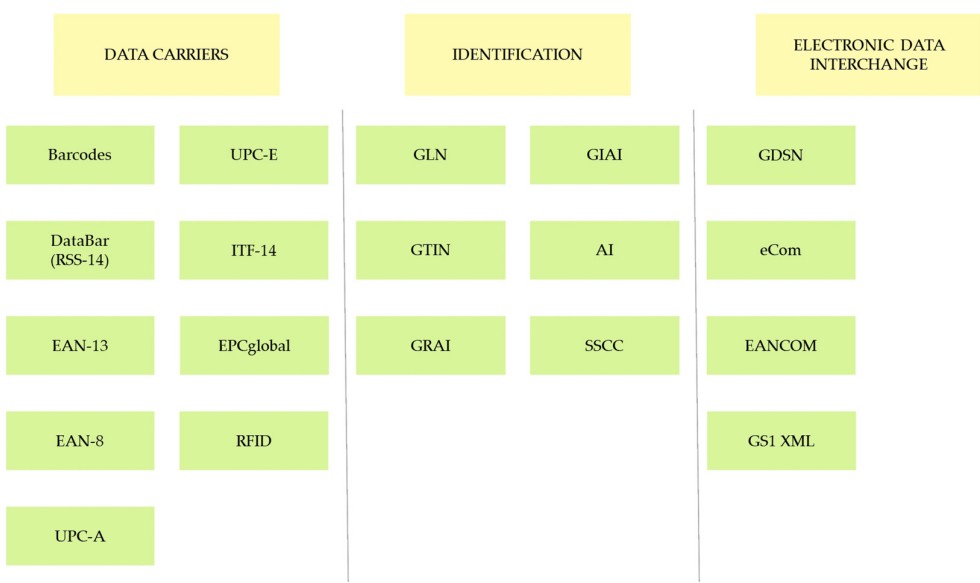

**Figure 1.** Different kinds of data carriers and identification keys.

Barcodes are one of the oldest machine-readable product master data carriers and are now widely used in various industries, following standards established by organisations like GS1. The GS1 was founded in 2005, by merging the European Numbering Association (EAN) and the Uniform Code Council (UCC), and today delivers the most prevalent standards. These standards enable efficient and safe operations in the supply chain by providing a universal language for product identification and information exchange. Adherence to these standards is essential for encoding data in global trade, while proper application of data carriers is crucial and follows a specific procedure [45].

Companies can assign identification numbers, such as the Global Trading Item Number (GTIN), to their products after obtaining a company prefix. There are different types of barcodes, depending on the type of item and the information they carry. The size and position of the barcode on the product should be determined based on the symbology and printing technology, while the quality of the barcode can be assessed using an ISO/IEC-based verifier [46]. The evolution of barcodes is making great progress, while future development will be based on RFID scanning technology [47], which enables the mass deployment of data carriers such as UHF EPC/RFID and HF EPC/RFID. The impact of RFID technology on the efficiency of logistics processes in retail supply chains has been analysed in [48].

The sources of product information may vary. It can be obtained directly from the manufacturer or the supplier. Once the product has received its unique identity and is prepared for distribution to the end user, it must be documented in the database of the

retailer and/or logistics company. To be included in the respective database, the product master data must be exchanged between the business partners. It is at this stage that most problems arise in connection with the product master data [15].

Enterprises have been exchanging bookings and orders with partners for a long time to better monitor the flow of goods [12]. Over time, various solutions have been developed and introduced to replace paper-based options and facilitate data exchange.

Nowadays, there are three common methods to receive or transmit product information electronically [2]: utilising an external electronic catalogue (e.g., GDSN), using an internal electronic catalogue and establishing a "direct linkage", or employing an extranet where suppliers manually input data. To promote the synchronisation and exchange of standardised product data within supply chains, external electronic catalogues are being created in different countries around the world following GDSN standards. Their main objective is to enable all trading partners to synchronise product data promptly and reliably via one connection [3,49]. Figure 2 illustrates the product information flows between the parties using an external electronic catalogue. The commencement of an item and entering corresponding product master information is executed at the source of the information (e.g., manufacturer, supplier). Subsequently, the process entails the recording of the product master details and enrollment in the vendor's data repository, enabling firms to exchange product data. The information transmission is concluded with the validation of receipt and dissemination of the company's data.

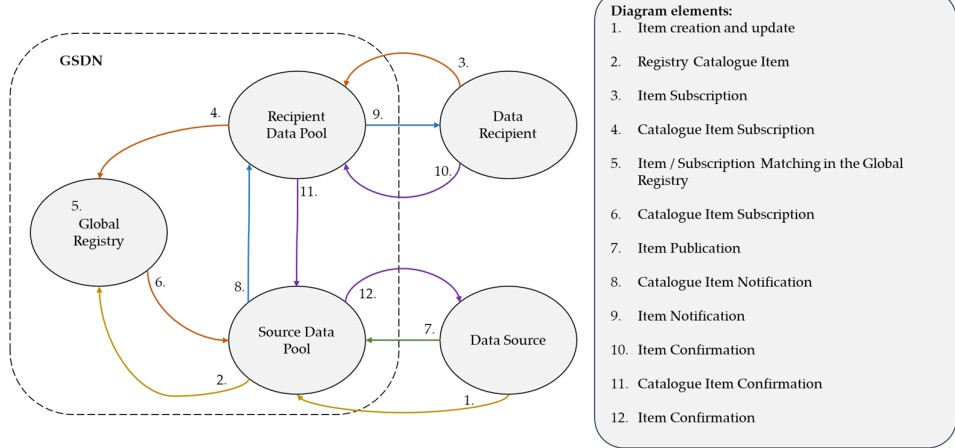

**Figure 2.** Information flows between the parties within the external electronic catalogue—GDSN.

Modern technologies have made managing large amounts of product data more challenging, especially across different organisational units, distribution, and supply channels [4]. The material master data, as a core functionality of the Enterprise Resource Planning (ERP) system, have become insufficient in dealing with numerous products from various sources. Trying to combine information from different sources can lead to increased resource consumption and errors, causing problems in deliveries, invoicing, inventory management, and transportation planning [4,50]. Increased awareness of problems regarding data management initiated Master Data Management (MDM) trends, in which a company's data are managed in a centralised manner, using a set of tools and processes for information systems standardisation (record data only once) and integration, and for data quality [50,51], whereas automated product classification is introduced to reduce human error [52,53]. These processes ensure data consistency and enable control of data usage for different operational and analytical applications.

In ensuring the optimal flow of goods through the entire supply chain, product master data must be enhanced with information necessary for logistic processes [51]: product serial numbers, such as GTIN, or another identification number; product name; product properties (e.g., material, hazardousness); product unit with unit of measure; supply units'

information; delivery units' information; supply filling and packing restrictions; handling, storage, and transport instructions; etc., manufacturer's information, delivery capability, and purchasing price. Accurate product logistic data are essential for designing and optimising logistic systems, determining storage and handling procedures, selecting transport modes, calculating transport capacities, determining packing strategies, scheduling cost-optimal orders, managing inventory, and allocating logistic resources [54].

## 4. Materials and Methods

### 4.1. Methodology

The approach used in the study involves the collection and examination of both primary and secondary data. Secondary data were obtained through a comprehensive review of previous research focussing on the challenges associated with incorrect or insufficient product information in logistics operations and an examination of the use of barcode technology. In parallel to the analysis of the existing literature, interviews were held with members of the GS1 Croatia organisation.

The primary data are acquired from research conducted on FMCG retailers located in Croatia, where 86.4% of the market is dominated by ten FMCG retailers [55]. This fact prompted us to focus on these companies, four of which agreed to participate in the study. Preliminary results of the study indicated the separation of two companies suitable for further, more comprehensive analysis. The selection criteria were based on the similarity of relationships within the information system in terms of the collection and dissemination of product master data, as well as the analogue use of product master data for the management of logistics processes. The selected companies have a market share of 35% [55], which we consider to be representative.

The data were collected in two ways: through semi-structured interviews and on-site screening. To thoroughly understand the complex processes involved in information exchange within the supply chain and its impact on logistic performance indicators, we conducted inquiries following the Who-Where-Why-How framework [56]. It was imperative to attain a thorough understanding of all the activities involved in the gathering and distribution of product master data, as well as the corresponding formal procedures (rules) set forth. To ensure accurate interpretation of the information, we sought confirmation through follow-up correspondence (e-mails, telephone conversations).

Following the identification of the study's range and level of intricacy, we embraced an approach grounded in process engineering and suggested the following research questions:

- The incoming control of the product and the associated master data reduces the possibility of delays in logistics process due to data errors.
- Product data errors have an impact on logistic process throughput time.

This entailed collecting qualitative and quantitative data, sampling the key performance indicators (KPI) of logistic processes, manipulating the data into appropriate formats, and generating flowcharts of the logistic processes through the utilisation of the ARIS 10.0 software.

We considered the process throughput time as the KPI for quantifying the impact of errors in the product master data on the process performance. Therefore, two inferential hypothesis testing procedures were conducted to analyse the statistical significance of the effect of product data master quality on the logistic process performance. Firstly, due to having one dichotomous (presence of inaccuracies) and one ratio scale variable (total throughput time in seconds), an independent sample $t$-test was employed to examine whether orders that were impacted by incorrect and insufficient product master data exhibited a higher overall throughput time in comparison to those with accurate data. Secondly, we had two variables measured on ratio scales (number of inaccuracies and total throughput time), so the Pearson correlation coefficient was used to investigate whether there is a positive relationship between the number of inaccuracies in the product master data and the overall throughput time. In other words, the question was whether a higher number of errors can be expected to be associated with a higher overall throughput time.

*4.2. Data Collection*

The first one, Company A, is the largest FMCG retailer in Croatia and employs over 10,000 people in a network of 700 points of sale, cross docks, and logistics centres. They sell around 35,000 different products through three distribution channels. The second one, Company B, is the leading distributor of global brands in beauty care, food, and non-food products. They manage more than 500 global and local brands, work with over 170 suppliers, and supply more than 80,000 customers.

Interviews were conducted with representatives from respective companies, and an analysis of data-sharing methodologies and product data distinctions was undertaken. Throughout multiple meetings with managers from the logistics division of both firms and engineers from the Information Technology (IT) departments, we collected data about the current practices and trends in the creation, collection, and dissemination of product master data. This included challenges faced by managers in merging and exchanging product master data within departments and with external parties (suppliers and buyers), as well as insights on acquisition, dissemination, and sales channels, and awareness of the significance of product master data in improving logistical operations.

First, it must be noted that, in practice, the administrative and physical acceptance of the product at the distribution centre does not have to take place at the same time, but it is necessary to check the condition of the goods. The administrative operations refer to the entry into the respective information system, while the physical reception refers to the storage in the distribution centre.

As per the criteria outlined in the methodology, Figure 3 illustrates the streamlined connections in the information system of the observed companies in terms of gathering and disseminating product master data. The commercial department records the initial product master data in the primary database (HOST) provided by the supplier. The quantity and type of information vary depending on the supplier. Product registration is typically performed manually, inputting only fundamental information (such as GTIN, product name, price, expiration date, lot number, and serial number) due to time and efficiency constraints.

The data stored in the HOST often require supplementation with additional information necessary for the execution of other activities, whether logistical or pertaining to e-commerce.

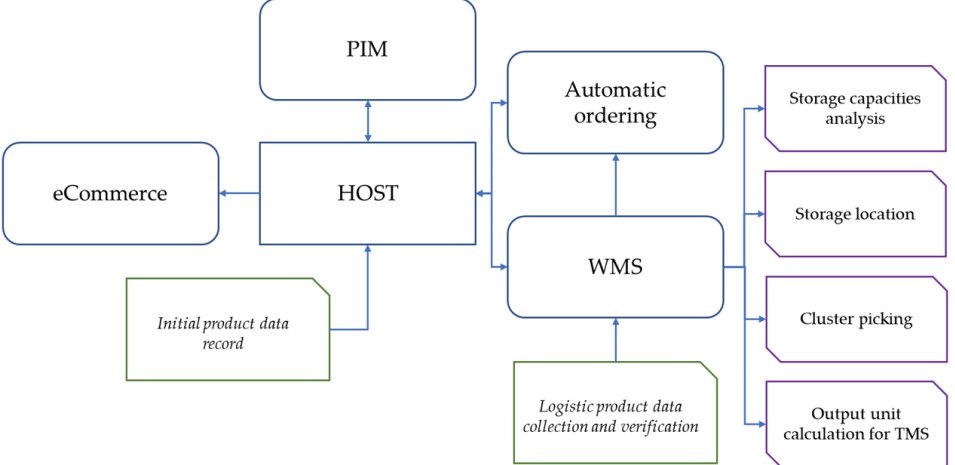

**Figure 3.** Simplified structure of the data-sharing system in observed companies.

The Product Information Management (PIM) system utilises product master data obtained from the HOST to establish connections between products and points of sale, while the eCommerce department imports and enhances them with the necessary attributes for the web shop.

In the execution of their activities, the logistics department relies on the WMS which utilises product data from the HOST as a foundation for analysing storage capacity, filling storage locations, cluster picking, and providing calculations to the Transport Management System (TMS). Product data enriched with details regarding size, weight, volume, and packaging type are crucial to facilitate logistical operations.

In Table 1, basic differences between respective companies are noted that can impact logistic process performance. Enrichment of product master data at Company A is initiated by the logistics department and executed by the commercial department, involving resources from multiple departments. The data are obtained from suppliers and manually added to the product database, which is a time- and resource-intensive task. On the other hand, Company B employs scanners for automated input control, which immediately updates the product master data with logistic information. However, this method requires additional warehouse space, personnel training, and financial investments, unlike Company A where only logistic resources are involved.

**Table 1.** Basic differences per parameters of interest.

| Parameters Observed | Company A | Company B |
|---|---|---|
| Business practice in acceptance of product | No input control | Input control by scanner |
| Trigger for enrichment in logistic product data (LPD) | Logistic department | Automated at input control |
| Enrichment of LPD made by | Commercial department employee | Input controller employee |
| Enrichment of LPD duration to be visible in WMS | Up to 1 month | Instantly or up to 1 h for small product |

For comparative analysis, we examined the order picking process, which is carried out similarly in both companies. We created a model of the process in the form of an event-driven process chain (EPC) as a baseline to measure the throughput time, which is the duration from when a picking order is released on the WMS to when the shipment is ready, as is outlined in Figure 4. We recorded the duration for each activity. Observed delays in the performance of any activities were separately measured, and the delays' reasons are noted.

The average number of shipping orders per day and the average number of picking orders per shipment (Table 2) were calculated using companies' data from a randomly selected period of three months. A total of 167 shipments in each company were randomly selected and monitored over one week. The share of monitored shipments for each company compared to its average weekly volume is shown in Table 2.

**Table 2.** Quantitative data of interest per company.

| Data | Company A | Company B |
|---|---|---|
| Average number of shipments per day | 140 | 700 |
| Average number of picking orders per shipment | 2.52 | 1.97 |
| Share of measured shipments compared to average weekly volume | 23.86% | 4.77% |

Once the orders underwent processing, including the picking and packing of items, they were transformed into shipments that were subsequently delivered to the customer. These shipments were either placed on homogenous or heterogenous pallets. The quantities of measurements conducted for each activity within the process can be found in Table 3. It should be noted that similar orders may require a different number of repeated steps (or activities) to complete the processing, resulting in different measurement requirements.

**Table 3.** Number of measurements per process activity.

| Process Activity | Company A Number of Measurements Performed | Company B Number of Measurements Performed |
|---|---|---|
| Taking an empty pallet | 89 | 94 |
| Going to the storage location navigated by WMS | 829 | 800 |
| Taking an item and scanning the barcode | 849 | 878 |
| Wrapping and labelling the pallet | 85 | 80 |
| Transporting the pallet to the shipping zone | 130 | 100 |

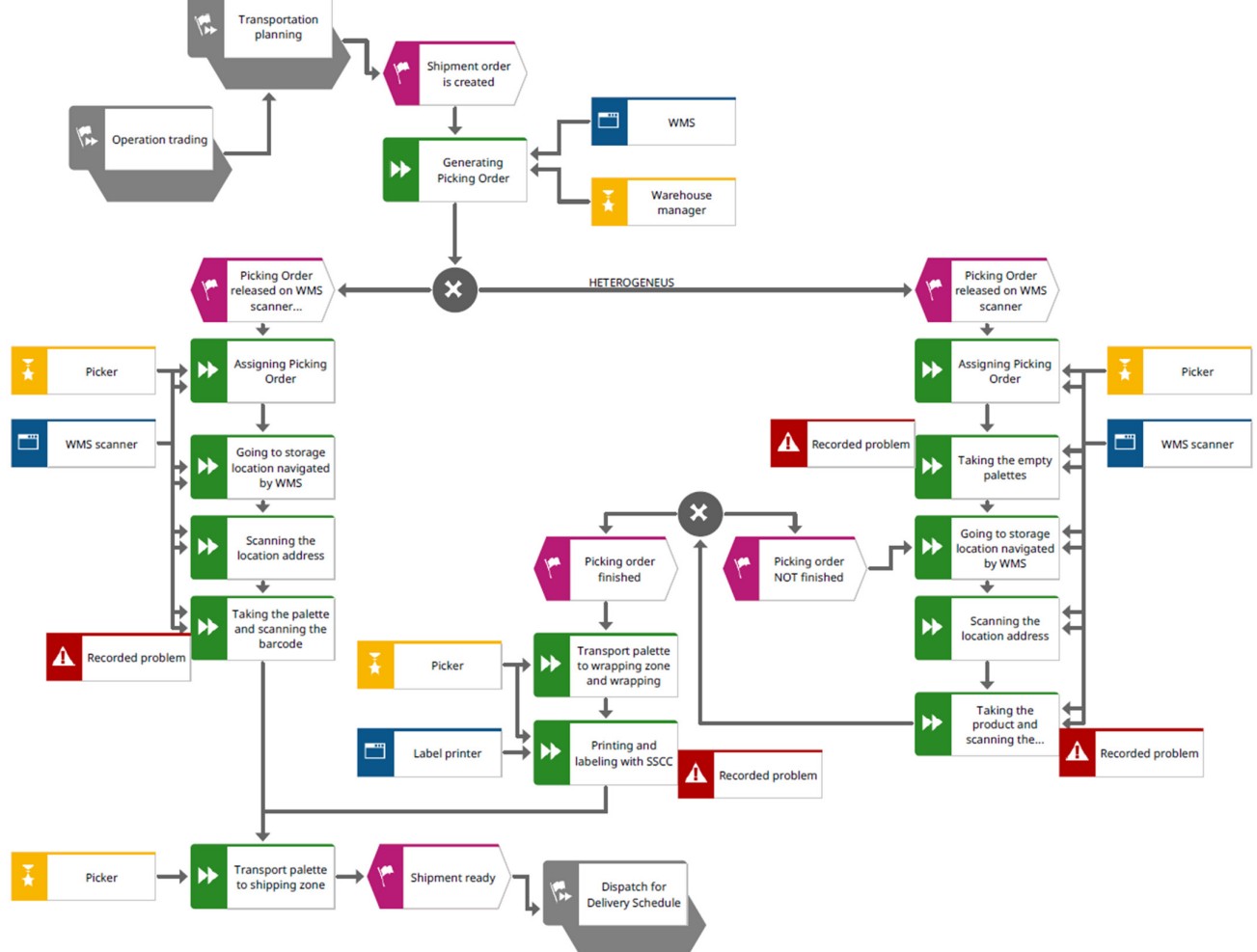

**Figure 4.** EPC diagram of the order picking process.

## 5. Results

Results from interviews on questions that relate directly to management's attitude to the accuracy of register product data are briefly summarised in Table 4.

To ensure comparability, from the monitored 167 shipments, we intentionally selected orders with similar characteristics in terms of type and quantity of items, totalling 55 orders for each company. A thorough review of the selected sample revealed that Company A had an average of 15.43 items per order, while Company B had a slightly higher average of 17.49 items per order.

**Table 4.** Results of the interview.

| Topic | Company A | Company B |
|---|---|---|
| Are the logistic managers aware of operational problems caused by inaccurate product data? | Yes | Yes |
| Whether they record such errors and analyse their consequences? | No | Partially |
| The quality of the product master data entered into the information system is evaluated: | Randomly, no standardised activities | Randomly for larger-dimensions products Targeted by brand for smaller-dimensions (cosmetics) products |
| If the product master data are insufficient, are outgoing orders dispatched or incoming orders accepted? | Yes | Yes |
| Are you familiar with GDSN? | Yes | Yes |
| Do you use GDSN? | No, only 1% of the product in the assortment included in the e-catalogue of the data pool | No, only 3% of the product in the assortment included in the e-catalogue of the data pool |

The results of the measurements performed in the field are graphically presented in Figures 5 and 6.

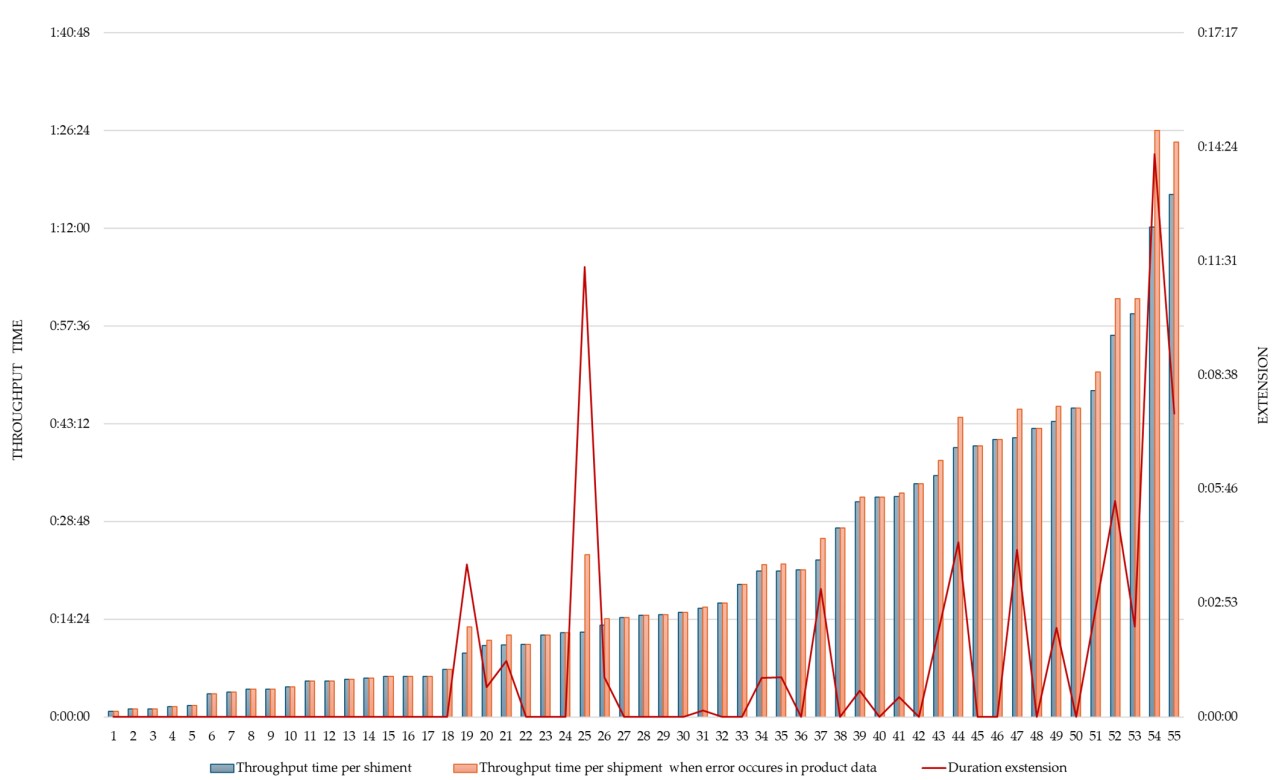

**Figure 5.** The order picking processes throughput time—Company A.

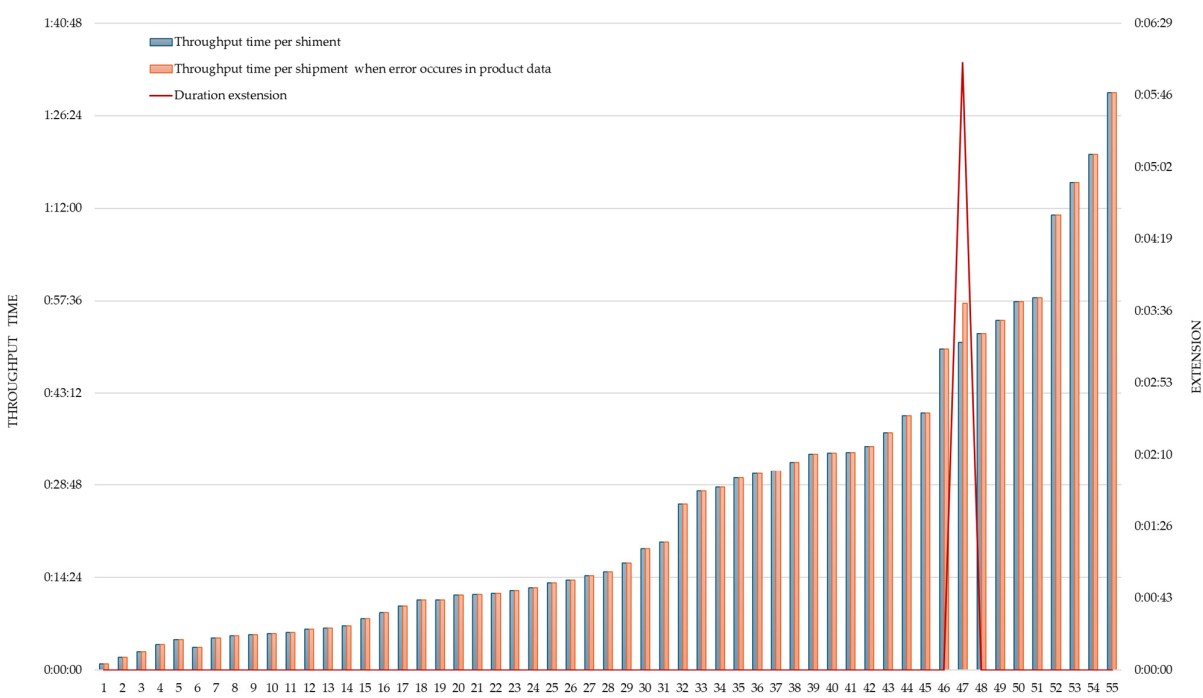

**Figure 6.** The order picking processes throughput time—Company B.

The average duration of the picking process per shipment at Company A, excluding any delays, was 21 min. Delays were observed in 20 orders, with some orders experiencing multiple errors, resulting in a total of 31 errors. These delays contributed to a 14.31% increase in the average processing time. Specifically, the average delay during the picking process was 3 min and 31 s. In contrast, only one error was found in the product master data at Company B, where a mandatory input check and data enrichment is performed at goods receipt. This single error caused a delay of 6 min and 5 s. Table 5 summarises the measurements obtained from both companies.

**Table 5.** Summary of the measurement results.

|  | Company A | Company B |
|---|---|---|
| Average throughput time [1] | 0:21:00 | 0:25:14 |
| Number of errors in the product master data | 31 | 1 |
| Number of orders with errors | 20 | 1 |
| Average duration of delays | 0:03:31 | 0:06:05 |
| Total delay | 1:10:24 | 0:06:05 |

[1] delay excluded.

We have found no statistically relevant correlation between the number of errors and the number of items per order. The number of errors and the number of items per order are graphically shown in Figures 7 and 8.

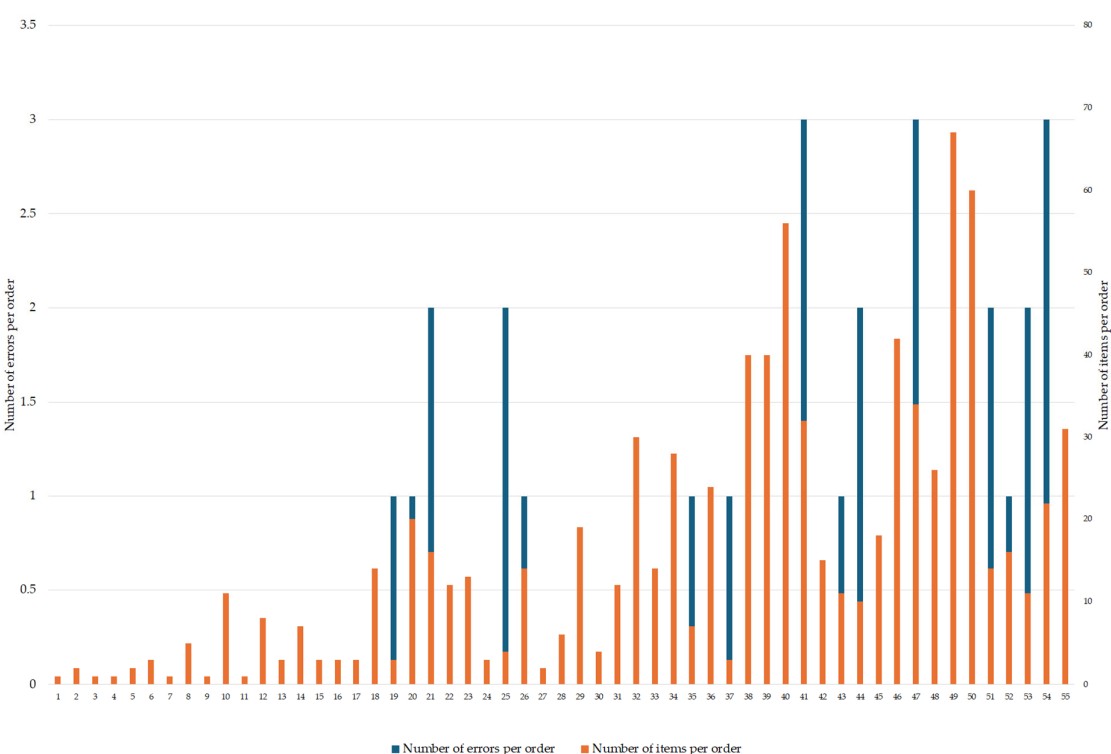

**Figure 7.** The number of errors in the product master data and the number of items per order—Company A.

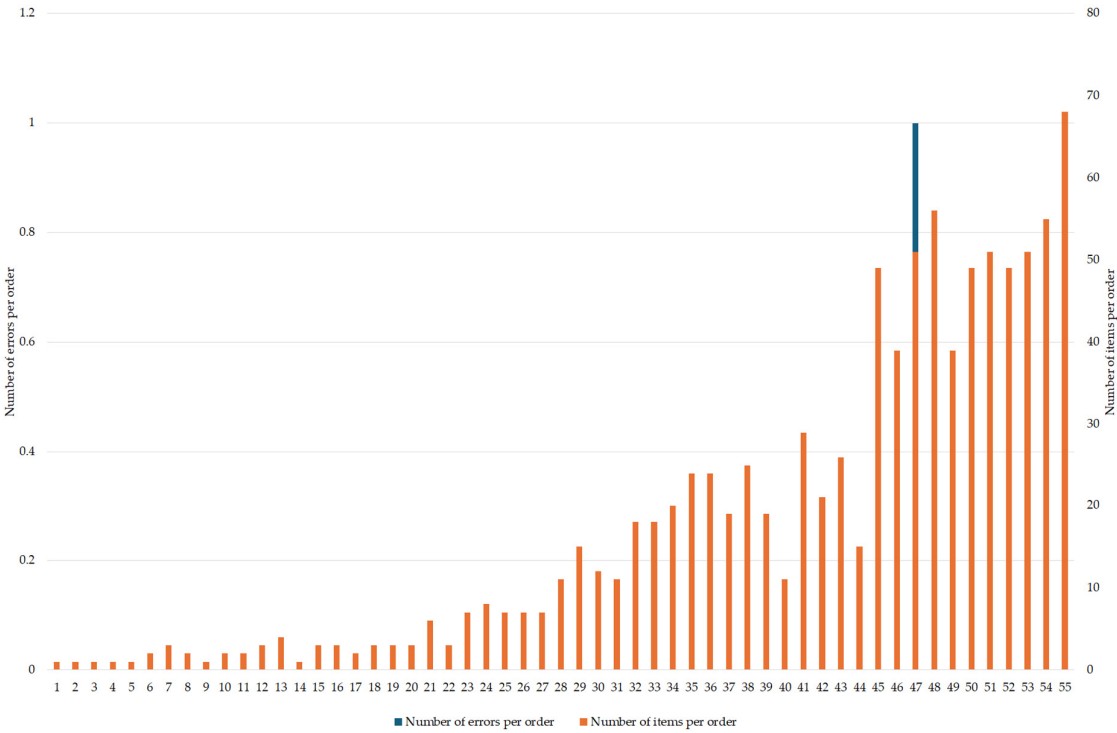

**Figure 8.** The number of errors in the product master data and the number of items per order—Company B.

The correlation between inaccuracies in the product data and the total throughput time was tested using the data from Company A, as in the selected sample only one error was recorded for Company B.

The descriptive statistics on overall throughput time across two samples based on product master data accuracy are presented in Table 6, while the results of the independent sample *t*-test are presented in Table 7.

**Table 6.** Descriptive statistics on overall throughput time across two samples based on product master data inaccuracy.

| Data Inaccuracy | *N* | *M* | *SD* | *SDM* |
|---|---|---|---|---|
| No | 35 | 839.57 | 804.75 | 136.03 |
| Yes | 20 | 2238.20 | 1366.22 | 305.50 |

**Table 7.** Results of the independent samples *t*-test with robust bootstrapped confidence intervals.

| | Levene's Test | | *t* Test | | | | | Bootstrapping Results | | |
|---|---|---|---|---|---|---|---|---|---|---|
| | | | | | | | | | BCa 95% CI | |
| | *F* | *p* | *t* | *df* | *p* | MD | $SE_{MD}$ | *p* | Lower | Upper |
| Equal variances assumed | 7.34 | 0.009 | −4.79 | 53 | 0.000 | −1398.63 | 333.76 | 0.001 | −2116.68 | −733.19 |
| Equal variances not assumed | | | −4.18 | 26.69 | 0.000 | −1398.63 | 333.76 | 0.002 | −2116.68 | −733.19 |

Note. Bootstrapping results are based on 1000 bootstrap samples. MD—Mean Difference, CI—Confidence Interval, BCa—Bias Corrected and Accelerated.

As we can see, orders that were affected with data inaccuracy had statistically significant higher overall throughput time than orders that had no data accuracy issues, on average.

Furthermore, there was also a statistically significant positive relationship between the number of accuracy errors and overall throughput time. In other words, the higher number of accuracy issues was related with higher overall throughput time (r = 0.59, *p* < 0.01, BCa CI [0.37–0.77]).

However, the somewhat smaller sample size used in the study warrants a more rigorous statistical approach to hypothesis testing, given that normality of the parameter sampling distribution cannot be assumed in smaller samples, and hence, we cannot rely on central limit theorem. This undermines the applicability of using parametric testing procedures, such as the *t*-test and Pearson correlation coefficient. Hence, distribution-free, non-parametric alternative procedures were also conducted. The results of both the Mann–Whitney test (z = −4.13, *p* < 0.001), as a non-parametric alternative to *t*-test, and Kendall's Tau rang correlation coefficient (τ = 0.47, *p* < 0.01, BCa CI [0.29–0.61]) were aligned with the aforementioned results of the *t*-test and Pearson correlation coefficient. Hence, only the results of parametric procedures are presented, given their higher statistical power. Furthermore, more robust bootstrap confidence intervals were calculated and presented. Overall, these results support the notion that the effect of product data master quality on the logistic process performance is existent in the population, that is, it has non-zero population value.

## 6. Discussion and Conclusions

In line with the initial aim of the study to assess management's understanding of the impact of product master data on the effectiveness of logistics processes, it can be deduced from the data collected and analysed that managers are aware of the complications arising from inaccurate product master data. However, they lack the motivation to automatically detect and analyse such inaccuracies. Instead, the assessment of product master data quality is sporadic, with corresponding activities lacking standardisation and automation. This aligns with the findings of [4,10,54], which reported that 81% of the 245 German companies

surveyed do not routinely employ standardised procedures for assessing the quality of product data.

Both companies prioritise adherence to the lead time as dictated by their business policies, which directly affects the flow of incoming and outgoing products. This means that incoming orders are received, and outgoing orders are shipped, even if the master data of the product are not entirely accurate or complete. Any necessary corrections are subsequently made internally.

Also, they are well acquainted with the Global Data Synchronization Network (GDSN); however, they do not perceive any advantages in employing the service. Within the electronic catalogue of the data pool, Company A can locate 1% of its product range, whereas Company B can locate 3%. These outcomes corroborate the findings of previous research [4–6] and provoke contemplation regarding the underlying reasons why manufacturers opt to refrain from inputting the master data of their products into electronic catalogues.

From production to sale to the end customer, products are transported (stored) in different loading units along the supply chain (e.g., in containers, pallets, and crates), which means that logistics-related product data vary at different stages of the movement. So, if logistics service providers were able to provide relevant product information to suppliers and retailers, it would be easier to exchange information and thus ensure the provision of appropriate product information. However, due to common limitations faced by logistics service providers in this regard, a significant portion of data is manually entered by various entities, leading to inconsistencies, errors, and inefficiencies. The conclusion of the study is in line with previous studies [2,6,10,16,32,37] and emphasises that the main obstacle to information exchange is the lack of compatibility between the databases of the different companies and their unwillingness to cooperate in the open exchange of information.

The planning and execution of logistics processes necessitate accurate and timely data from the involved companies; therefore, proper quality of the product master data is a condition sine qua non. Nevertheless, the potential benefits of advanced information systems and technologies in managing logistics are limited by the quality of the product master data.

The results of the second research objective, which relate to the measurement of the influence of master data on the efficiency of logistics processes, confirm the assumption of a significant influence.

Our research reveals that despite the implementation of input control (Company B) using a scanner, errors in logistic product data persist. The number of errors detected (see Table 5) speaks in favour of the implementation of input controls by the scanner, but the disadvantage is reflected in the limitations of the scanner in terms of the minimum and maximum dimensions of the product. In the observed case, the logistical product data in the database must be manually checked and corrected for products with small dimensions, which account for a significant turnover in the respective company.

The lack of a significant correlation between errors in product data and order items (see Figures 7 and 8) suggests that errors occur randomly when matching product codes and data records in the local database, especially during data exchange within the supply chain. The problems encountered during the evaluation of shipments in each company analysed, which can be attributed to incorrect or inadequate logistical data, can be briefly described as inaccurately stocked items, delays in replenishment and search for specific items, repetitive packing and handling activities, and the need to allocate time and resources to correct errors. Problematic activities in the observed process are marked with an exclamation mark in Figure 4.

The results of the 55 shipments selected from the 167 shipments measured show an average increase of up to 14.31% in the time required for order consolidation when errors occur in the product data. The average process throughput time at Company A, exclusive of any delays, is comparatively shorter due to a distinct warehouse layout. Additionally, Company A manages a smaller volume of items per order, contributing to the reduced

duration. Consequently, the average delay in resolving errors is also shorter at Company A. Nevertheless, we observed a higher occurrence of errors in the product master data at Company A, which significantly increased the overall delay in the order picking process in comparison to Company B. These product master data errors not only resulted in delays during activities such as item retrieval and barcode scanning but also prolonged the time taken for activities like retrieving an empty pallet or navigating to the storage location facilitated by the WMS system. This prolongation can be attributed to factors such as incorrect dimensions or weight, improper storage location, or inadequate quantity of items at the picking level. Furthermore, it also extended the time required for SSCC (Serial Shipping Container Code) printing due to incorrect information on the label.

The answer to the question of whether a higher number of errors can be expected to be associated with a higher total throughput time was influenced by the results of the statistical significance tests, the *t*-test, and Pearson's correlation coefficient. For orders affected by inaccurate data, the process throughput time was significantly higher compared to orders without data accuracy issues. A positive correlation was also found between the number of accuracy errors and the process throughput time. However, due to the small sample size used in the study, a more rigorous statistical approach was required. Therefore, non-parametric alternative methods were also performed, which led to analogous results as the parametric methods.

The research results confirm the importance of high-quality product master data in the supply chain, especially for the flow of materials and information, and that this has a significant impact on the performance of logistics processes, including delays in order picking due to late, inaccurate, and incomplete data. In addition, insufficient quality of product master data hinders the optimisation of logistics processes and the introduction of advanced technologies such as augmented reality, automated storage and retrieval systems (ASRS), and automated guided vehicles (AGVs).

The research addresses a significant gap in the field of logistics by empirically measuring the impact of product master data quality on the performance of logistics processes. It contributes to the academic debates on the digital transformation of supply chains, emphasising the crucial role of data quality in logistics operations. Through a novel methodological approach that combines measuring process activity duration with identifying causes using inferential statistical tests, a unique framework is proposed for evaluating the efficiency of logistics processes. This methodology could be adopted or further developed in future studies. Despite the focus on the FMCG sector, the insights provided regarding the importance of product data accuracy can be extrapolated to diverse industries, fostering interdisciplinary research on data management in supply chain contexts.

From a practical perspective, this study looks at management strategy and emphasises the urgent need for companies to prioritise improving the quality of product data as part of their logistics strategy. The study shows that investment in technologies such as automated data capture scanners, while initially resource-intensive, can improve data quality and reduce errors over time. This has the potential to influence decisions on budget allocation and technology adoption. In addition, the study shows a clear link between data quality and operational performance, the pursuit of operational excellence. Companies can use these findings to support the adoption of standardised protocols and quality checks for product data. It also highlights the disadvantages of insufficient data synchronisation between supply chain partners, which could prompt companies to work more closely with their partners on data management. The study emphasises that high-quality product master data are not only a regulatory necessity, but also a strategic advantage that can provide a competitive edge by improving logistics performance.

These implications can serve as a foundation for both further research and the strategic development of logistics operations in various industries.

Although the case study was carefully conducted, in the final stage it was limited to two Croatian companies in the FMCG distribution and retail industry, however representative for the Croatian market. A broader generalisation of the results would require

a larger sample (including some other markets) and consideration of additional logistics processes and the possibilities of incorporating additional logistics information into the product master data. These areas should be the focus of future research, as well as the relationship between the quality of product master data and resource consumption in logistics processes. The current business practises in distribution centres that implemented advanced technologies should be considered too.

**Author Contributions:** Conceptualization, D.B. and R.S.; methodology, D.B.; software, D.B.; validation, D.B. and A.I.; formal analysis, M.Ž., D.B. and A.I.; investigation, D.B. and M.Ž.; resources, D.B.; data curation, D.B., M.Ž. and A.I.; writing—original draft preparation, D.B. and M.Ž.; writing—review and editing, R.S.; visualization, D.B., M.Ž. and A.I.; supervision, R.S.; project administration, D.B.; funding acquisition, D.B. All authors have read and agreed to the published version of the manuscript.

**Funding:** This work was funded by the Fundamental Research Funds for the University of Zagreb (Grant number 210244-ZUID 2021/2022). The APC was funded by the University of Zagreb, Faculty of Transport and Traffic Sciences.

**Data Availability Statement:** The data presented in this study are available on request from the corresponding author. The data are not publicly available due to observed companies' policy.

**Acknowledgments:** We are grateful to the management and the personnel of the companies who enabled us to carry out the case study. Many thanks to GS1 Croatia for providing the data carrier knowledge.

**Conflicts of Interest:** The authors declare no conflicts of interest.

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
