# Peer review of "Impact of the Product Master Data Quality on the Logistics Process Performance"

_logistics, 2024_

Round 1

Reviewer 1 Report

Comments and Suggestions for Authors

It is quite a comprehensive experiment. Can you introduce the flow? and also can you introduce the implications, in practical and academic ways?

Section 2 should be a literature review.

Can you provide a literature review regarding the latest developments?

Can you elaborate Figure 1? And, some definitions and explanations of data carrier are needed.

Can you also explain Figure 2? It is expected to discuss how low quality it is recently to show the research gap.

At the beginning of Section 3. It is expected there is a whole methodology of your studies. For examples, it should be started from the company selection mechanism. Also, what kind of comparison they are going to do in this sections. Otherwise, it can be easily being challenged as there are just two samples. Also, in Section 4, it is expected some framework of what results are presenting, and how it is presented.

Finally, it is advised that if you can provide some implications academically and practically.

Author Response

Dear Reviewer 1,

We have revised the manuscript according to  comments. Please find the details of our responses  (blue-coloured text). All revisions to the manuscript are highlighted in blue.

  1. It is quite a comprehensive experiment. Can you introduce the flow? and also can you introduce the implications, in practical and academic ways?

The flow of the study is introduced with additional clarifications in the methodology, see Section 4, from line 228.

The conclusions on the academic and practical implications have been added in Section 6, lines 520 – 542.

  1. Section 2 should be a literature review.

Section 2. A literature review has been added, accordingly.

  1. Can you provide a literature review regarding the latest developments?

The literature review regarding the latest developments has been provided in Section 2., and more recent research on this topic is also included.

Also, the latest developments are added in lines 66-69.

  1. Can you elaborate Figure 1? And, some definitions and explanations of data carrier are needed.

The explanation has been provided in lines 141 – 157, and short definitions of data carriers are added. Figure 1 presents the variability of carriers and identification keys in short.

  1. Can you also explain Figure 2? It is expected to discuss how low quality it is recently to show the research gap.
    The explanation has been provided in lines 194 – 200.

Explanation:

Section 3 which we called “Demystification of product master data and logistic data” was added as a short overview of how the known system works. To date, networks and systems have been created that enable the exchange of product data, but there are still problems with the accuracy and completeness of the data. With the globalization of the market, these problems are becoming ever greater. To gain insight into the current state of data exchange, we spoke a lot with GS1 representatives and various IT experts. Figure 2 was created based on discussions with GS1 Croatia representatives and findings from other external product catalogs (e.g. Icecat).

  1. At the beginning of Section 3. It is expected there is a whole methodology of your studies. For examples, it should be started from the company selection mechanism. Also, what kind of comparison they are going to do in this sections. Otherwise, it can be easily being challenged as there are just two samples. Also, in Section 4, it is expected some framework of what results are presenting, and how it is presented.

The Materials and Methods (Section 4 in the current version) has been restructured and updated accordingly.

The framework of what the results present is provided in section 5.

  1. Finally, it is advised that if you can provide some implications academically and practically.

The conclusion on the academic and practical implications has been added in Section 6, lines 520 – 542.

Reviewer 2 Report

Comments and Suggestions for Authors

Expressing gratitude for the chance to review a manuscript titled "Impact of the product master data quality on the logistics process performance," I extend congratulations to the authors for their commendable study. The paper delves into the oversight of product data quality's significance in the context of performing logistics processes. This topic holds substantial practical and scientific implications for the supply chain and logistics domain. However, it is essential for the authors to address several concerns that I believe warrant attention.

Please clarify what the acronym FMCG means in the abstract?

The Methodology section should be prepared in higher quality, there is lack the arguments in a solid and critical way why these methods were chosen. I found it difficult to follow the authors' logic reading the methodology part. I recommend to provide more detailed information about interview, what type of interview was used, a structure of interview questions. Doubts arise, on what basis is Figure 3 prepared, where does the information come from?

In my opinion, it should be explained more precisely how the order picking processes were measured (Figure 5 and 6).

The text (i.e., line 302) must be reviewed and corrections made.

Author Response

Dear Reviewer 2,

We have revised the manuscript according to the comments. Please find the details of our responses. All revisions to the manuscript are highlighted in blue.

Expressing gratitude for the chance to review a manuscript titled "Impact of the product master data quality on the logistics process performance," I extend congratulations to the authors for their commendable study. The paper delves into the oversight of product data quality's significance in the context of performing logistics processes. This topic holds substantial practical and scientific implications for the supply chain and logistics domain. However, it is essential for the authors to address several concerns that I believe warrant attention.

  1. Please clarify what the acronym FMCG means in the abstract?

The acronym FMCG has been explained accordingly.

  1. The Methodology section should be prepared in higher quality, there is lack the arguments in a solid and critical way why these methods were chosen. I found it difficult to follow the authors' logic reading the methodology part. I recommend to provide more detailed information about interview, what type of interview was used, a structure of interview questions. Doubts arise, on what basis is Figure 3 prepared, where does the information come from?

In my opinion, it should be explained more precisely how the order picking processes were measured (Figure 5 and 6).

The text (i.e., line 302) must be reviewed and corrections made.

The Materials and Methods (Section 4 in the current version) has been restructured and updated accordingly.

Figure 3 has been prepared by the authors, based on the data and information obtained in the case study, as explained in Section 4. Additional information is given from 291-298.

Description and explanation of the order picking process measurements are explained in lines 341 - 355.

The text in the former line 302 is corrected. See line 330, 340.

Reviewer 3 Report

Comments and Suggestions for Authors

REVIEW COMMENTS

Ref: logistics-2893918

Title : “Impact of the product master data quality on the logistics process performance”

The authors have made an attempt to a good work however the following comments must be addressed before its next steps.

1.     In abstract, FMCG must be expanded as “Fast-moving consumer goods” for its first time usage.

2.     The term “chronographic measurements of the logistics processes” could be cited with appropriate reference

3.     In line 10-11, “The statistical significance of the impact was determined using the t-test for independent samples and Pearson's correlation coefficient and additionally validated by non-parametric alternative methods.”---------It is better to brief the motivation behind choosing these methods for the proposed work.

4.     In line, 115-116 the objectives given as

“to assess management awareness of the importance of product data quality at Croatian logistics industry and

to quantify its influence on the performance of logistics processes.”----------------

5.     There is an ambiguity in the second objective. The authors should make the proposed objective very clear.

6.     How did they select “Croatian logistics industry” for the proposed study has not been clearly explained.

7.     In line 293, “event-driven process chain (EPC) model”--------the authors are suggested to give proper citation for the model.

8.     In line 295, the term “WMS” must be expanded for its first time usage.

9.     In line 299, Figure 4. “EPC diagram of the order picking process.”--------The letters must be legible.

10.  Have the authors made any “hypothesis testing”. If so the hypothesis must be clearly stated before the test.

11.  In line 382, “non- parametric alternative procedures were also conducted.”-------------these

12.  In general, to attract researchers by the proposed work, the number of questions and the categories in the Questionnaire must be improved to a larger extent, also the number of hypothesis should be increased thereby  the comparative analysis must be made with an appropriate statistical tools.

Author Response

Dear Reviewer 3,

We have revised the manuscript according to the comments. Please find the details of our responses. All revisions to the manuscript are highlighted in blue.

The authors have made an attempt to a good work however the following comments must be addressed before its next steps.

  1. In abstract, FMCG must be expanded as “Fast-moving consumer goods” for its first time usage.

The acronym FMCG has been explained accordingly.

  1. The term “chronographic measurements of the logistics processes” could be cited with appropriate reference

The term “chronographic measurements of the logistics processes” is replaced with the expression “process activities were timed using a stopwatch“.   

  1. In line 10-11, “The statistical significance of the impact was determined using the t-test for independent samples and Pearson's correlation coefficient and additionally validated by non-parametric alternative methods.”---------It is better to brief the motivation behind choosing these methods for the proposed work.

We have updated the text accordingly (line 11-12).

Explanation:

The types of inferential procedures are determined by the variable and measurement scale type that was used. Since we had one dichotomous (independent - the presence of inaccuracies (yes/no), and one ratio scale (dependent - total throughput time in seconds) we used the most common inferential procedure - independent t-test. In another case, we had two variables measured on ratio scales (number of inaccuracies and total throughput time), so the Person correlation coefficient was used since it is the most common procedure for two ratio scales. The reasoning for the choice of inferential procedures is not commonly discussed in scientific papers unless is something out of the ordinary, so we didn’t mention that in the paper also. The reasoning behind using both parametric and non-parametric procedures is already clearly explained in the paper, along with the usage of more robust bootstrap confidence intervals.

  1. In line, 115-116 the objectives given as

“to assess management awareness of the importance of product data quality at Croatian logistics industry and

to quantify its influence on the performance of logistics processes.”----------------

We do not understand this comment.

  1. There is an ambiguity in the second objective. The authors should make the proposed objective very clear.

We clarified the objectives of the research accordingly. See lines 80-83.

  1. How did they select “Croatian logistics industry” for the proposed study has not been clearly explained.

In Section 4, the procedure for selecting the company was added. See lines 234-242.

  1. In line 293, “event-driven process chain (EPC) model”--------the authors are suggested to give proper citation for the model.

In lines 330-331 (in the last version was 293) we changed the text to “We created a model of the process in the form of Event-driven process chain (EPC) as a baseline to measure the throughput time”

  1. In line 295, the term “WMS” must be expanded for its first time usage.

The acronym WMS has been explained accordingly.

  1. In line 299, Figure 4. “EPC diagram of the order picking process.”--------The letters must be legible.

Figure 4 has been improved.

  1. Have the authors made any “hypothesis testing”. If so the hypothesis must be clearly stated before the test.

The research questions are added accordingly. See lines 252-256.

  1. In line 382, “non- parametric alternative procedures were also conducted.”-------------these

The reasons for applying non-parametric methods are explained in lines 416 onwards.

  1. In general, to attract researchers by the proposed work, the number of questions and the categories in the Questionnaire must be improved to a larger extent, also the number of hypothesis should be increased thereby the comparative analysis must be made with an appropriate statistical tools.

The authors agree that the aims and scopes of this study could have been broader. However, this study had a narrower focus on exploring the basic effect of product data master quality on the logistic process. This shortcoming is now mentioned in the discussion section.